# Enhancing Recommendation Accuracy and Diversity with Box Embedding: A Universal Framework

## ABSTRACT

Recommender systems have emerged as an indispensable mean to meet personalized interests of users and alleviate information overload. Despite the great success, accuracy-oriented recommendation models are creating information cocoons, i.e., it is becoming increasingly difficult for users to see other items they might be interested in. Although recent studies start paying attention to enhancing recommendation diversity, models based on point embedding fail to describe the range of user preferences and item features well, which is essential for diversified matching. To this end, we propose LCD-UC, a novel recommendation framework based on box embedding to improve recommendation diversity with the recommendation accuracy maintained. Specifically, LCD-UC creates hypercubes to represent users and items using box embedding for high model flexibility and expressiveness. Then, a hypercube similarity scoring function is designed to measure the similarity between hypercubes representing users and items. To make a balance between the accuracy and diversity of recommendations and achieve personalized diversity needs, we further develop a user-item pairwise attention mechanism as well as a user uncertainty masking mechanism in LCD-UC. Besides, we present two new metrics for better evaluation on recommendation diversity, which address the issue that existing metrics only consider the coverage of categories while ignore the frequency of categories. The extensive experiments on three real-world datasets show that LCD-UC can improve both recommendation accuracy and diversity over three base models, and is superior to six state-of-the-art recommendation models. An online 10-day AB test also demonstrates that LCD-UC can improve the performance of a real-world advertising system.

## CCS CONCEPTS

• **Computing methodologies** → **Machine learning**.

## KEYWORDS

Recommender System, Box Embedding, Graph Neural Networks

**ACM Reference Format:**
Anonymous Author(s). 2023. Enhancing Recommendation Accuracy and Diversity with Box Embedding: A Universal Framework. In *Proceedings of The Web Conference 2024 (WWW '24)*. ACM, New York, NY, USA, 10 pages. https://doi.org/XXXXXXX.XXXXXXX

## 1 INTRODUCTION

In recent years, recommender systems have become ubiquitous across a wide range of applications, including location-based services [3], e-commerce platforms [15] and online video websites [6]. A dominant approach to building a recommendation model, the kernel of a recommender system, is Collaborative Filtering (CF), which evolves from traditional Matrix Factorization algorithms [19, 24] to innovative Deep Neural Network architectures such as Autoencoders [42] and Graph Neural Networks (GNNs) [17, 49].

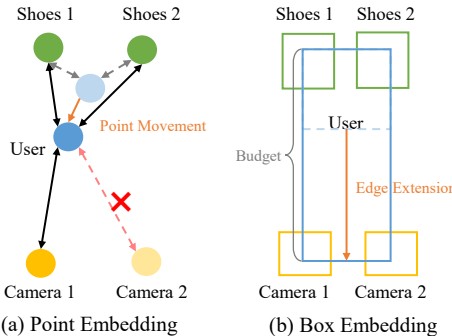

(a) Point Embedding      (b) Box Embedding

**Figure 1: The Advantage of Box Embedding w.r.t. flexibility and expressiveness. Consider a scenario where a user has had interactions with Shoe 1 and Shoe 2, and has a new interaction with Camera 1. (a) Using point embedding, the new interaction should be modeled by moving the points, but the movement of the user node towards the Camera 1 node is constrained by the two shoe nodes, making it impossible to recall Camera 2. (b) With box embedding, simply extending the edges of the box can maintain the user's original interest and match Camera 2 to increase recommendation diversity. Moreover, if the vertical axis represents the price, the vertical edge of the user box can represent the range of the budget.**

While CF solutions are highly effective in suggesting items meeting personal interest and successfully alleviating information overload [51], an excessive pursuit of personalized recommendations can result in the creation of information cocoons [25], i.e., it is difficult for users to encounter other content that might pique their interest. To address this issue, several latest studies pay attention to recommending accurate and diverse items for users [13, 46, 53, 58].

Nevertheless, existing methods for diversified recommendation are mostly based on point embeddings, which are incapable of modeling the range of user preferences and item features, resulting in low model flexibility and expressiveness. As illustrated in Figure 1(a), point embedding methods require point movement to capture the use's new interaction with Camera 1, but the movement is constrained by the shoe nodes. Thus, the model fails to recall Camera 2. To address this issue, we propose to use box embedding [48]

to encode users and items into hypercubes (see Figure 1(b)). With the edges describing the range of user preferences and item features, these hypercubes can accurately represent and calculate the correlation between users and items. Moreover, compared with point embedding models, such range representation in box embedding models can enhance recommendation diversity, since recalling items that meet a certain range typically results in a greater number of items than recalling those that meet a certain value.

However, learning box embeddings to improve recommendation diversity is not a trivial task, and there are two major challenges: **How to measure the similarity between hypercubes of users and items.** Most box embedding models calculate the size of the intersecting volume as the similarity between hypercubes [12, 33]. While intuitive, this method can easily lead to a similarity measure of zero due to non-intersection in one dimension, creating difficulty in box embeddings learning in high-dimensional scenarios. In addition, Zhang et al. [57] explore a distance function to measure the distance between user hypercuboids and item points, which provides improvements in terms of flexibility and model expressiveness. However, item representations based on point embeddings sacrifice the expressive power of box embeddings on the item side, which is not optimal. Thus, a new hypercube similarity scoring function is needed.

**How to make a balance between recommendation accuracy and diversity.** Although enhancing recommendation diversity can help break through information cocoons, indiscriminately promoting diversity may lead to the recommendation of items that users are not interested in, thus doing harm to the accuracy of recommendations. Furthermore, different users have different needs for diversity, and some highly specialized items are only suitable for a small proportion of users. Thus, it's necessary to consider such personalized and specific diversity requirements when making recommendations for better balance between accuracy and diversity.

To tackle the above challenges, this paper proposes the **L**ist-**C**heck-**D**ecide framework with the **UnC**ertainty masking mechanism (LCD-UC) to enhance both recommendation accuracy and diversity. Specifically, the LCD framework first creates hypercubes for users and items based on the point embedding generated by a base model. Then, a novel hypercube similarity scoring function is designed to measure the relevance between users and items. To meet personalized diversity needs, we further develop a user-item pairwise attention mechanism as well as a user uncertainty masking mechanism for users. Consequently, LCD-UC is capable of learning flexible and expressive representations of users and items, thereby enhancing the effectiveness of recommendations.

Besides, existing metrics for evaluating recommendation diversity are not particularly apt, and the reasons are two-fold. On the one hand, some studies use metrics for evaluating item novelty and popularity (like SRDP) as a reflection of item diversity [46]. These metrics cannot completely substitute for diversity metrics, as novel and unpopular items do not necessarily belong to different categories with popular ones. On the other hand, metrics like *Genre Coverage* [28, 38] that consider item categories merely assess the coverage of categories, while the frequency of categories remains unconsidered. For instance, consider the following two sets of video recommendations: **1) Nine sports videos and one car video**, and **2) Five sports videos and five car videos**. Both recommendation results cover the sports and car categories. However, we believe that result 2 demonstrates greater diversity than result 1.

To this end, we propose two new evaluation metrics for recommendation diversity that consider the frequency of items from different categories.

The main contributions of this work are highlighted as follows:

- We develop LCD, a universal framework to encode user and item representations into hypercubes, and make recommendations base on a novel hypercube similarity scoring function. We also propose the uncertainty masking mechanism for personalized diverse recommendation (Section 3).
- We analyze the limitations of existing diversity evaluation metrics w.r.t. item category frequency, and propose two new metrics namely ICSI and ILCS for better measurement of recommendation diversity (Section 4).
- We conduct comprehensive experiments to show that LCD-UC can improve both recommendation accuracy and diversity over three base models and achieve state-of-the-art performances compared to six baseline models. We also conduct an AB test to show the effectiveness of LCD-UC online (Section 5).

## 2 RELATED WORK

In this section, we review related work relevant to this study, including diversified recommendation and box embedding.

### 2.1 Diversified Recommendation

The diversity of recommendation can be viewed from aggregate or individual perspectives. Methods improving aggregate diversity aim to recommend items from as many categories as possible to all users [2, 23, 34, 36, 55], while the goal for individual diversity based recommendation is to achieve a balance between accuracy and diversity for each target user [54]. This paper focuses on individual diversity, which can mainly be divided into three categories. The first category adopts post-processing heuristics [4, 5, 39]. For example, Carbonell and Goldstein [5] proposed Maximal Marginal Relevance (MMR) to selectively choose an item with the highest local combination of similarity score to the query and dissimilarity score to previously ranked documents. The second category leverages the determinantal point process (DPP) to measure set diversity by describing the probability for all subsets of the item set, i.e., assigning higher probability to sets of items that are diverse from each other [7, 13, 47, 53]. The third category models diversified recommendation as end-to-end supervised learning task and optimizes both diversity and accuracy through a single model [10, 46, 58].

In this paper, instead of designing a dedicated model, we propose a universal framework to grant diversified recommendation ability to existing end-to-end recommender systems.

### 2.2 Box Embedding

Box embedding [48], also known as hypercube representations [8] or hypercuboid representation [57], are useful abstractions to express high-order information of the data. It has attracted attention in the field of machine learning [11, 26, 31, 45, 48, 50, 56], and has been used in diverse applications, such as knowledge bases [1, 9, 29, 35, 37, 41] and image embedding [40].

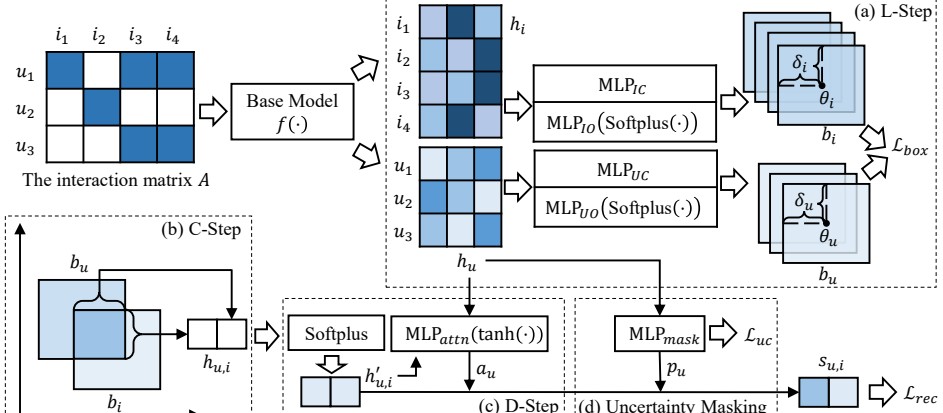

**Figure 2: The architecture of LCD-UC. LCD-UC consists of the LCD framework and the uncertainty masking mechanism to learn box embedding for users and items. There are three steps in LCD, namely the L-Step, the C-Step and the D-Step.**

Recently, box embedding is introduced to recommender systems [8, 12, 27, 32, 33, 57]. Mei et al. [32] proposed to embed users and items in high dimensional latent space, with a simplified variant of box embedding. Zhang et al. [57] proposed to represent the user as a multi-dimensional hypercuboid while the items were represented as points. The edges of the user hypercuboid is used to describe the ranges of preferences, which enhances the representation capacity in capturing the diversity of preferences. Mei et al. [33] proposed to use probabilistic box embeddings for effective and efficient ranking, in which queries and items are parameterized as high-dimensional axis-aligned hyper-rectangles. Liang et al. [27] modeled the recommendation as a logical reasoning task and embedded each query as a box rather than a single point in the vector space, which was able to model sets of users or items enclosed and logical operators over boxes in a more natural manner.

In this paper, we propose a new similarity scoring function for box embedding to model the personalized uncertainty of users' preferences for diversified recommendation.

## 3 METHODS

In this section, we first present the overview of the proposed LCD-UC framework (see Figure 2), and then delve into the details of its underlying framework LCD and the uncertainty masking mechanism. Finally, we propose the optimization approach of LCD-UC.

### 3.1 Overview

Formally, we focus on the standard setting where there are a set of users $\mathcal{U}$, a set of items $\mathcal{I}$, a set of categories $C$ and an interaction matrix $A \in \mathbb{R}^{|\mathcal{U}| \times |\mathcal{I}|}$. Each item $i \in \mathcal{I}$ is associated with a set of categories $C_i \subseteq C$. Each entry $A_{u,i} = 1$ in $A$ denotes that there exists implicit feedback such as clicks, likes and forwards between user $u \in \mathcal{U}$ and item $i \in \mathcal{I}$, while $A_{u,i} = 0$ indicates no observed feedback. Then, the goal of recommendation is to generate a set of recommendable items $\mathcal{R}_u$ (usually the top-$k$ items ranking by a recommendation model) for each user $u$.

We propose the LCD-UC framework to make recommendation. As shown in Figure 2, LCD-UC is built upon *The LCD Framework*

by introducing *The Uncertainty Masking Mechanism*. Specifically, there are three steps in the LCD framework, including the L-Step, the C-Step and the D-Step. Intuitively, the L-Step uses box embeddings to list the range of user preferences and item features in different aspects (dimensions). The C-Step checks how the item matches the user in each dimension. The D-Step then calculates the comprehensive matching score and decides whether the user is interested in the item. To enhance recommendation diversity, the uncertainty masking mechanism learns personalized masks to decrease the importance of user preferences in some aspects, thus recalling more items.

### 3.2 The LCD Framework

The LCD framework is a universal framework for enhancing recommendation diversity, which can transform the vector representations of users and items outputted by underlying base models into high-dimensional box embeddings and calculate their similarity. The base models can use matrix factorization–models [24] or graph-based models [17, 49], and so on.

*3.2.1 L-Step: List.* Given the user embeddings $\{h_u | u \in \mathcal{U}\}$ and the item embeddings $\{h_i | i \in \mathcal{I}\}$ generated by a base model, the aim of L-Step is to identify what user preferences are and how many aspects of these preferences an item meets. These preference needs are often manifested in terms of ranges. For example, in an e-commerce scenario, users have a budget range and items have a price fluctuation range. Hence, we use box embeddings to model such preference range. Specifically, we use different multi-layer perceptrons (MLPs) to learn the centers and offsets of the user and item boxes. The formulas are as follows:

$$\theta_u = \mathrm{MLP}_{UC}(h_u); \quad \delta_u = \mathrm{Softplus}(\mathrm{MLP}_{UO}(h_u)), \tag{1}$$

$$\theta_i = \mathrm{MLP}_{IC}(h_i); \quad \delta_i = \mathrm{Softplus}(\mathrm{MLP}_{IO}(h_i)), \tag{2}$$

$$h_u, h_i \in \mathbb{R}^{d' \times 1}; \quad \theta_u, \delta_u, \theta_i, \delta_i \in \mathbb{R}^{d \times 1}. \tag{3}$$

Here, $\theta_u$ and $\delta_u$ (or $\theta_i$ and $\delta_i$) denote the box center and the box offset of user $u$ (or item $i$), respectively. $d'$ is the vector dimension outputted by the base model, and $d$ is the vector dimension of the

box embedding. $\text{Softplus}(x) = \log(1 + e^x)$ is an activation function to make sure that the box offsets are positive.

Then, the box embeddings of $u$ and $i$ are denoted as:

$$b_u = \text{Box}(h_u) = (\theta_u, \delta_u); \quad b_i = \text{Box}(h_i) = (\theta_i, \delta_i). \quad (4)$$

Each dimension in $b_u$ (or $b_i$) represents a preference, and $b_u$ (or $b_i$) as a whole can be regarded as a preference list.

*3.2.2 C-Step: Check.* The C-Step is designed to check whether an item is suitable for a user. More precisely, we compute the dimension-wise matching degree according to the user and item preference lists $b_u$ and $b_i$ derived from the L-Step as follows:

$$h_{u,i} = |b_u \wedge b_i| = \{|b_u^{(0)} \wedge b_i^{(0)}|, \cdots, |b_u^{(d)} \wedge b_i^{(d)}|\}. \quad (5)$$

Here, $|b_u^{(k)} \wedge b_i^{(k)}|$ is the length of the intersection between box $b_u$ and $b_i$ w.r.t. dimension $k (k = 1, \ldots, d)$, which is computed as:

$$|b_u^{(k)} \wedge b_i^{(k)}| = \max(0, \min(\top_u^{(k)}, \top_i^{(k)}) - \max(\bot_u^{(k)}, \bot_i^{(k)})), \quad (6)$$

where $\top_u^{(k)}$ and $\bot_u^{(k)}$ (or $\top_i^{(k)}$ and $\bot_i^{(k)}$) are the upper and lower bounds of $b_u^{(k)}$ (or $b_i^{(k)}$), respectively, and are formulated as follows:

$$\top_u^{(k)} = \theta_u^{(k)} + \delta_u^{(k)}; \quad \bot_u^{(k)} = \theta_u^{(k)} - \delta_u^{(k)}, \quad (7)$$

$$\top_i^{(k)} = \theta_i^{(k)} + \delta_i^{(k)}; \quad \bot_i^{(k)} = \theta_i^{(k)} - \delta_i^{(k)}. \quad (8)$$

Although intuitive, optimizing box embeddings according to Eq. (6) using gradient-based training methods can be difficult when two boxes do not intersect, as the gradients in relation to this training pair would be zero. To address this issue, we follow Dasgupta et al. [11] and leverage independent Gumbel distribution to model the parameters of box embeddings. Specifically, given the temperature parameter of Gumbel distribution $\beta$, we have:

$$\min(b_u^{(k)} \wedge b_i^{(k)}) \sim \text{Gumbel}(-\beta \ln(e^{-\frac{\bot_u^{(k)}}{\beta}} + e^{-\frac{\bot_i^{(k)}}{\beta}}), \beta), \quad (9)$$

$$\max(b_u^{(k)} \wedge b_i^{(k)}) \sim \text{Gumbel}(\beta \ln(e^{\frac{\top_u^{(k)}}{\beta}} + e^{\frac{\top_i^{(k)}}{\beta}}), \beta). \quad (10)$$

Then, the lower and upper bounds of the interaction length w.r.t. $u$ and $i$ at dimension $k$ are the expectation of $\min(b_u^{(k)} \wedge b_i^{(k)})$ and $\max(b_u^{(k)} \wedge b_i^{(k)})$, respectively, which are computed as:

$$\bot_{u,i}^{(k)} := E(\min(b_u^{(k)} \wedge b_i^{(k)})) = -\beta \text{LogSumExp}(-\frac{\bot_u^{(k)}}{\beta}, -\frac{\bot_i^{(k)}}{\beta}), \quad (11)$$

$$\top_{u,i}^{(k)} := E(\max(b_u^{(k)} \wedge b_i^{(k)})) = \beta \text{LogSumExp}(\frac{\top_u^{(k)}}{\beta}, \frac{\top_i^{(k)}}{\beta}), \quad (12)$$

where $\text{LogSumExp}(x, y) = \log(e^x + e^y)$.

Finally, the $|b_u^{(k)} \wedge b_i^{(k)}|$ based on Gumbel distribution is calculated as:

$$|b_u^{(k)} \wedge b_i^{(k)}| = \beta \log(1 + \exp(\frac{\top_{u,i}^{(k)} - \bot_{u,i}^{(k)}}{\beta} - 2\gamma)), \quad (13)$$

where $\gamma$ is the Euler-Mascheroni constant.

*3.2.3 D-Step: Decide.* The goal of D-Step is to determine whether item $i$ satisfies user $u$ according to the dimension-wise matching degree $h_{u,i}$. A naive idea is to sum up all the dimension of $h_{u,i}$. However, this approach fails to take the personalized diversity needs of users w.r.t. different items into consideration. Therefore, we devise a hypercube similarity scoring function $s(u, i)$ based on

a user-item pairwise attention mechanism to perform weighted averaging on $h_{u,i}$ as follows:

$$h'_{u,i} = \text{Softplus}(h_{u,i}), \quad (14)$$

$$a_u = \text{MLP}_{attn}(\tanh(h_u \cdot h'_{u,i})), \quad (15)$$

$$s(u, i) = a_u^\top h'_{u,i}, \quad (16)$$

where $a_u$ is the attention weight vector encoding the preference of both user $u$ and item $i$. Note that here we use $h'_{u,i}$ instead of $h_{u,i}$, because the Softplus activation function in Eq. (14) can introduce a log operation, such that the sum operation on $h'_{u,i}$ is equivalent to the multiplication operation on the transformation of $h_{u,i}$. This multiplication operation reflects the classical volume calculation concept in box embeddings to some extent, and can also avoid training difficulties caused by a zero volume due to non-intersection.

## 3.3 The Uncertainty Masking Mechanism

The intention of the uncertainty masking mechanism is to break through information cocoons by reducing the importance of some user preferences, thereby increasing the number of items that match with the user. As each dimension in $h'_{u,i}$ represents the matching degree between $u$ and $i$ in different aspects, we can use a 0/1 mask vector $p_u$ to disable certain dimensions. Considering that the importance of preferences in different dimensions varies for users, we develop a personalized masking mechanism by associating each dimension of $p_u$ with a random variable $p_u^{(k)} \sim \text{Bernoulli}(w_u^{(k)})$. Here, $w_u^{(k)} \in [0, 1]$ is the Bernoulli weight parameterized by the original user embedding $h_u$ as follows:

$$w_u = \text{MLP}_{mask}(h_u). \quad (17)$$

Then, the hypercube similarity scoring function in Eq. (16) is modified to incorporate the uncertainty masking mechanism as:

$$s(u, i) = a_u^\top (p_u \cdot h'_{u,i}). \quad (18)$$

To train our model in an end-to-end fashion, we utilize the Gumbel-Max reparametrization trick [22, 30] to relax the discrete $p_u^{(k)}$ into a continuous variable in $[0, 1]$. Specifically, given $\delta \sim \text{Uniform}(0, 1)$, we have:

$$p_u^{(k)} = \sigma((\log \delta - \log(1 - \delta) + w_u^{(k)})/\tau), \quad (19)$$

where $\sigma(x) = \frac{1}{1+e^{-x}}$ is the sigmoid function and $\tau$ is the temperature hyper-parameter. As $\tau \to 0$, $p_u^{(k)}$ gets closer to being binary.

## 3.4 Model Optimization

In this subsection, we present the loss function for training LCD-UC and analyze the time complexity.

*3.4.1 Training Loss.* LCD-UC is trained by three losses, including *the recommendation loss*, *the box regularization loss* and *the mask regularization loss*.

**Recommendation Loss.** Follow He et al. [17], we use the *Bayesian Personalized Ranking (BPR) Loss* as the recommendation loss. The intuition is to maximize the similarity score between positive user-item pairs (i.e., $A_{u,i} = 1$), while minimize the similarity between negative ones (i.e., $A_{u,i} = 0$). Formally, the BPR loss is defined as:

$$\mathcal{L}_{rec} = -\frac{1}{|O|} \sum_{(u,i,j) \in O} \log \sigma(s(u, i) - s(u, j)). \quad (20)$$

where $O \subseteq \{(u, i, j) | u \in \mathcal{U}, i, j \in \mathcal{I}, A_{u,i} = 1, A_{u,j} = 0\}$ is the training data.

**Box Regularization Loss.** In box embeddings, the size of the box is an important factor affecting the effect of the model. Bigger boxes cover more space and are often overlapped with more boxes, which is beneficial to improving recommendation diversity. However, oversized boxes can make it difficult to create distance between dissimilar boxes, decreasing the recommendation accuracy. To this end, we devise a box regularization loss to restrict the box size as follows:

$$\mathcal{L}_{box} = \frac{1}{|\mathcal{V}|} \sum_{v \in \mathcal{V}} | \sum_{k=1}^{d} (\delta_v^{(k)} - \eta_{box})|, \tag{21}$$

where $\mathcal{V} = \mathcal{U} \cup \mathcal{I}$ is the set of all users and items, and $\eta_{box} \geq 0$ is a hyper-parameter. Generally, a larger $\eta_{box}$ results in bigger boxes and greater diversity.

**Mask Regularization Loss.** In LCD-UC, another factor that affects the recommendation accuracy and diversity is the Bernoulli parameters $w_u^{(k)}$ in the uncertainty masking mechanism. Smaller $(w_u^{(k)})$s generate mask vectors with more dimensions valued at zero, i.e., mask more aspects of preferences for greater diversity. To control the size of $w_u^{(k)}$, we design the following mask regularization loss:

$$\mathcal{L}_{uc} = \frac{1}{|\mathcal{U}|} \sum_{u \in \mathcal{U}} | \sum_{k=1}^{d} (w_u^{(k)} - \eta_{uc})|, \tag{22}$$

where $\eta_{uc} \in [0, 1]$ is a hyper-parameter.

Finally, the global training loss is the combination of the above three losses:

$$\mathcal{L} = \mathcal{L}_{rec} + \lambda_{box} \mathcal{L}_{box} + \lambda_{uc} \mathcal{L}_{uc}, \tag{23}$$

where $\lambda_{box}$ and $\lambda_{uc}$ are hyper-parameters to control the influences of $\mathcal{L}_{box}$ and $\mathcal{L}_{uc}$, respectively.

*3.4.2 Time Complexity Analysis.* In this part, we analyze the additional time complexity introduced by LCD-UC on top of the base model. In L-Step, the time complexity to create box embeddings for all users and items is $O((|\mathcal{U}| + |\mathcal{I}|) \cdot d'd)$. The time complexity to compute $h_{u,i}$ in C-Step for a user-item pair is $O(d)$, and the time complexity of the hypercube similarity scoring function with attention mechanism is $O(d^2)$. In the uncertainty masking mechanism, the time complexity to mask $h'_{u,i}$ is $O(d^2)$. Finally, to train LCD-UC, the time complexities of $\mathcal{L}_{rec}$, $\mathcal{L}_{box}$ and $\mathcal{L}_{uc}$ are $O(|O|(d + d^2))$, $O((|\mathcal{U}| + |\mathcal{I}|)d)$ and $O(|\mathcal{U}|d)$, respectively. Thus, the overall time complexity to train LCD-UC is $O((|\mathcal{U}| + |\mathcal{I}|) \cdot d'd + |O|d^2)$. As $d'$ and $d$ are small constants, the overall time complexity is linear to the number of all users and items $|\mathcal{U}| + |\mathcal{I}|$ and the size of training set $|O|$, showing the efficiency of LCD-UC.

# 4 TWO DIVERSITY METRICS

In this section, we formulate the two proposed metrics which aim to evaluate the diversity of recommendations based on item categories. Specifically, we propose the *Item Category's Simpson's Index (ICSI)* metric and the *Intra-List Category Similarity (ILCS)* metric for macro and micro diversity evaluation, respectively.

**Item Category's Simpson's Index (ICSI)** measures the macro diversity based on the *Simpson's Index* [44]. ICSI takes into account both the number of categories present, and the frequency of each category. ICSI is calculated using the following formulas:

$$ICSI = \frac{1}{|\mathcal{U}|} \sum_{u \in \mathcal{U}} (1 - \sum_{c \in C} p_{u,c}^2), \tag{24}$$

$$p_{u,c} = \frac{f_{u,c}}{\sum_{c' \in C} f_{u,c'}}, \tag{25}$$

$$f_{u,c} = \sum_{i \in \mathcal{R}_u} \sum_{c' \in C_i} \mathcal{X}(c = c'), \tag{26}$$

where $\mathcal{X}(\cdot)$ is the indicative function, and $p_{u,c}$ is the frequency of category $c$ appearing in all recommended items for user $u$. ICSI reflects the probability that two items randomly chosen from a list of recommendations belong to different categories. The larger the value of ICSI, the higher the diversity of the recommendation results. The range of ICSI is $[0, 1 - \frac{1}{|C|}]$, where ICSI is 0 when all the items belong to one category, and ICSI is $(1 - \frac{1}{|C|})$ when the frequency of each category is the same. Note that we do not scale the range of ICSI to [0,1], and thus we can directly compare new calculated values with existing results when the number of item categories increases.

**Example 1.** Consider the two sets of video recommendations mentioned in Section 1: 1) Nine sports videos and one car video, and 2) Five sports videos and five car videos. For result 1, the ICSI is $1 - 0.9^2 - 0.1^2 = 0.18$, while for result 2, the ICSI is $1 - 0.5^2 - 0.5^2 = 0.5$. Thus, result 2 demonstrates greater diversity than result 1.

**Intra-List Category Similarity (ILCS)** measures the micro diversity of $\mathcal{R}_u$ by the mean *Jaccard Index* [21] between all pairs of items in $\mathcal{R}_u$ for each user $u$, and then takes the average. It is formulated as follows:

$$ILCS = \frac{1}{|\mathcal{U}|} \sum_{u \in \mathcal{U}} \frac{1}{|\mathcal{R}_u|(|\mathcal{R}_u| - 1)} \sum_{(i,j) \in \mathcal{R}_u} \frac{|C_i \cap C_j|}{|C_i \cup C_j|}. \tag{27}$$

As ILCS denotes the mean pairwise similarity between recommended items, a smaller ILCS value is supposed to indicate greater diversity. The range of ILCS is $[0, 1]$, where ILCS is 0 when none of the items share the same category, and ILCS is 1 when all the items belong to the same category set.

**Example 2.** Consider the recommendation result $\mathcal{R}_u = \{1, 2, 3\}$ for user $u$. When

Case 1 $C_1 = \{$Sports, Car$\}, C_2 = \{$Sports, Game$\}, C_3 = \{$Car, Game$\}$, the ILCS value is $\frac{1}{6} \times (\frac{1}{3} + \frac{1}{3} + \frac{1}{3} + \frac{1}{3} + \frac{1}{3} + \frac{1}{3}) = \frac{1}{3}$;

Case 2 $C_1 = \{$Sports$\}, C_2 = \{$Game$\}, C_3 = \{$Car$\}$, the ILCS value is $\frac{1}{6} \times (\frac{0}{2} + \frac{0}{2} + \frac{0}{2} + \frac{0}{2} + \frac{0}{2} + \frac{0}{2}) = 0$;

Case 3 $C_1 = \{$Sports$\}, C_2 = \{$Game, Shooting$\}, C_3 = \{$Car$\}$, the ILCS value is $\frac{1}{6} \times (\frac{0}{3} + \frac{0}{2} + \frac{0}{3} + \frac{0}{3} + \frac{0}{2} + \frac{0}{3}) = 0$.

Note that the ICSI metric and the ILCS metric focus on different aspects of diversity, and they cannot replace each other. Specifically, in Example 2, Case 1 and Case 2 have different ILCS values. However, the ICSI values of these two cases are both $1 - (\frac{1}{3}^2 + \frac{1}{3}^2 + \frac{1}{3}^2) = \frac{2}{3}$. In contrast, Case 2 and Case 3 have the same ILCS value. Nevertheless, the ICSI of Case 2 is $\frac{2}{3}$ while for Case 3, the ICSI value is $1 - (\frac{1}{4}^2 + \frac{1}{4}^2 + \frac{1}{4}^2 + \frac{1}{4}^2) = \frac{3}{4}$.

**Comparison with Existing Diversity Metrics.** *Genre Coverage (GC)* and *Intra-List Distance (ILD)* are popular metrics to assess the diversity of the recommended set [28, 38]. Similar to ICSI, GC is a macro diversity metric, and it counts the number of relevant

**Table 1: The statistics of datasets.**

| Datasets | Interactions | # Users | # Items | % Density |
|---|---|---|---|---|
| MovieLens | 998,971 | 6,040 | 3,706 | 4.463 |
| KuaiRec | 14,977,539 | 7,176 | 10,728 | 19.455 |
| MIND | 76,978,213 | 872,083 | 130,379 | 0.068 |

genres (i.e., categories) recommended to the user. As mentioned in Section 1, GC fails to distinguish the two sets of video recommendations in Example 1, since they all cover the sports and car categories. ILD is a micro diversity metric, and it measures the diversity of the set of recommended items by the mean *Hamming Distance* between category vectors. Comparing to the mean *Jaccard Index* in ILCS, ILD only takes the differences of item categories into consideration, while ignoring the commonalities, resulting in sub-optimal evaluation.

## 5 EXPERIMENTS

In this section, we conduct extensive experiments and answer the following research questions:

- **RQ1**: How can LCD-UC improve the recommendation performance?
- **RQ2**: How does LCD-UC perform compared to existing recommendation methods?
- **RQ3**: How do the proposed mechanisms in LCD-UC take effect?
- **RQ4**: How do different settings influence the effectiveness of LCD-UC?
- **RQ5**: Can LCD-UC learn interpretable box embeddings?
- **RQ6**: Can LCD-UC improve the performance of real-world recommender systems.

### 5.1 Experimental Settings

*5.1.1 Datasets.* We evaluate the performance of LCD-UC on three real-world datasets with different scales and densities, namely MovieLens [16], KuaiRec [14] and MIND [52]. We summarize the statistics of these datasets in Table 1. The detailed information of these datasets is listed as follows.

- **MovieLens** [16] contains the anonymous ratings of 3,706 movies made by 6,040 MovieLens users who joined MovieLens in 2,000. We construct the interaction matrix $A$ with $A_{u,i} = 1$ if the rating for movie $i$ by user $u$ is not less than 4.0, otherwise, $A_{u,i} = 0$. For each user, we sort the items by the interaction time and split 70%, 10% and 20% items as the training, validation and test sets, respectively.
- **KuaiRec** [14] is a dense dataset derived from the recommendation logs of Kuaishou, a popular video-sharing mobile application. Following the instruction of Gao et al. [14], we create the binary interaction matrix $A$ by setting $A_{u,i} = 1$ if the watch ratio of $u$ w.r.t. $i$ is not less than 2.0, otherwise, $A_{u,1} = 0$. KuaiRec contains a big matrix and a small matrix. We use the big matrix as the training data, split 10% of the small matrix as the validation data and leave the rest of the small matrix as the test data.
- **MIND** [52] is a large-scale dataset collected from anonymized behavior logs of Microsoft News website for news recommendation research. It contains impression logs with both positive and negative feedback on news from users. We use the positive

feedback as the 1 entries in the interaction matrix $A$, and follow the original training/validation/test split in MIND.

*5.1.2 Baseline Methods.* We evaluate LCD-UC with nine state-of-the-art baseline methods, including MF [24], NCF [18], GRU4REC [20], PD-GAN [53], NGCF [49], LightGCN [17], DGCN [58], HCUR [57] and CBox4CR [27]. The details of the baseline methods are listed as follows.

- **MF** [24], or Matrix Factorization, is a traditional method used in collaborative filtering. It learns the underlying factors by employing the alternating least squares technique.
- **NCF** [18] is a neural collaborative filtering method that combines multi-layer perceptron with generalized matrix factorization to capture non-linearities.
- **GRU4REC** [20] leverage the GRU network to model user interaction sequence for session-based recommendation.
- **PD-GAN** [53] suggests the adoption of an adversarial learning process to understand user's individual preferences and item diversity. It leverages the DPP model as a generator to yield diverse results.
- **NGCF** [49] is a message passing architecture that aggregates information across the user-item interaction graph. This approach is designed to exploit high-order connection relationships.
- **LGCN** [17] (short for LightGCN) is a light-weight GCN, which is easy to train and has good generalization ability.
- **DGCN** [58] performs rebalanced neighbor discovering, category-boosted negative sampling and adversarial learning on top of GCN to improve recommendation diversity.
- **HCUR** [57] models user interests as a hypercuboid instead of a point in the space, and learns the recommendation score by calculating a compositional distance between the user hypercuboid and the item.
- **CBox4CR** [27] combines contrastive learning with collaborative reasoning to learn the distinctive box representations for the user's preference and the logical query base on the historical interaction sequence.

*5.1.3 Parameter Settings.* We implement LCD-UC with Pytorch and the model is optimized using the Adam optimizer with learning rate 0.001 during the training phase. By default, $d'$ and $d$ are set to 64, $\beta$ is set to 0.05, $\tau$ is set to 0.1, $\eta_{box}$ is set to 0.4, $\eta_{uc}$ is set to 0.8, $\lambda_{box}, \lambda uc$ are set to 1. For all the baseline methods, we tune the parameters according to the validation set and report the best results. All the experiments are conducted on a machine with 256GB memory using a single NVIDIA GeForce RTX 3090 GPU.

*5.1.4 Metrics.* We use the proposed ICSI and ILCS metrics to evaluate the recommendation diversity. To evaluate recommendation accuracy, we adopt two popular evaluation measures [43] including Recall@$k$ and Normalized Discounted Cumulative Gain (NDCG@k). Recall@$k$ refers to the percentage of relevant items that are found within the top-$k$ recommendations. NDCG@$k$ evaluates the quality of ranking, taking into account the position of correctly recommended items. For each user in MovieLens and KuaiRec, we rank all the items (exclude items the user has interacted with), while for the user in MIND, we rank the items given by the test set. Then, we select the highest ranked $k$ items for recommendation. Note that

**Table 2: The effectiveness of LCD-UC. The best results are illustrated in bold.**

| Dataset | MovieLens | | | | KuaiRec | | | | MIND | | | | Rank ↓ |
|---|---|---|---|---|---|---|---|---|---|---|---|---|---|
| Metric | NDCG ↑ | Recall ↑ | ILCS ↓ | ICSI ↑ | NDCG ↑ | Recall ↑ | ILCS ↓ | ICSI ↑ | NDCG ↑ | Recall ↑ | ILCS ↓ | ICSI ↑ | |
| MF | 0.0308 | 0.0258 | 0.3271 | 0.8764 | 0.0325 | 0.0042 | 0.6168 | 0.6559 | 0.9511 | 0.7528 | 0.5474 | 0.0955 | 2.00 |
| + LCD-UC | **0.0772** | **0.0677** | **0.1751** | **0.8992** | **0.0596** | **0.0130** | **0.2616** | **0.8413** | **0.9781** | **0.7625** | **0.5332** | **0.1206** | **1.00** |
| NGCF | 0.0533 | 0.0459 | 0.2293 | 0.8330 | 0.1924 | 0.0596 | 0.1757 | 0.8015 | 0.9741 | 0.7614 | 0.5620 | 0.0728 | 2.00 |
| + LCD-UC | **0.1065** | **0.1027** | **0.2252** | **0.8577** | **0.2632** | **0.0846** | **0.1173** | **0.8626** | **0.9781** | **0.7626** | **0.5612** | **0.0742** | **1.00** |
| LGCN | 0.0759 | 0.0738 | 0.2531 | 0.8235 | 0.2147 | 0.0610 | 0.1542 | 0.8212 | 0.9760 | 0.7611 | 0.5635 | 0.0704 | 2.00 |
| + LCD-UC | **0.1091** | **0.1071** | **0.2498** | **0.8399** | **0.2708** | **0.0894** | **0.1111** | **0.8668** | **0.9787** | **0.7625** | **0.5631** | **0.0718** | **1.00** |

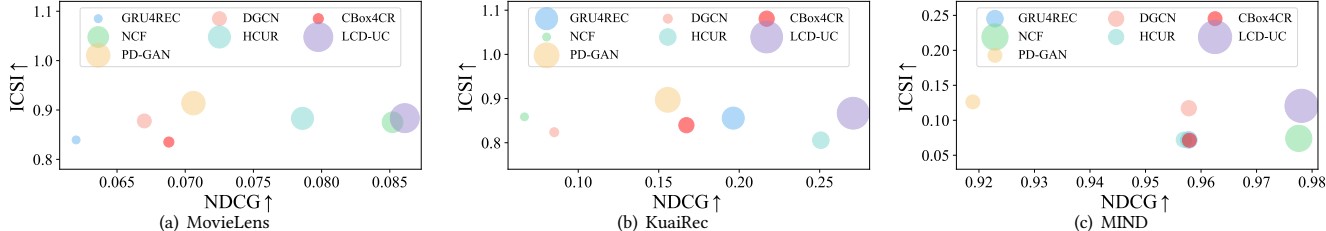

(a) MovieLens    (b) KuaiRec    (c) MIND

**Figure 3: The experimental results of overall performance. The sizes of the circles indicate the average rank of different methods, i.e., the better rank, the larger size. The NDCG value and the ICSI value represent recommendation accuracy and diversity, respectively, and the circles further to the right and top indicate better performances.**

**Table 3: Ablation Study. The best results are illustrated in bold and the number underlined is the runner-up.**

| Dataset | | MovieLens | | | | KuaiRec | | | | MIND | | | | Rank ↓ |
|---|---|---|---|---|---|---|---|---|---|---|---|---|---|---|
| Metric | | NDCG ↑ | Recall ↑ | ILCS ↓ | ICSI ↑ | NDCG ↑ | Recall ↑ | ILCS ↓ | ICSI ↑ | NDCG ↑ | Recall ↑ | ILCS ↓ | ICSI ↑ | |
| MF | Box | 0.0703 | 0.0617 | 0.1760 | 0.8958 | 0.0588 | 0.0094 | 0.3921 | 0.8236 | 0.9747 | 0.7604 | 0.5412 | 0.1082 | 2.58 |
| | LCD | 0.0732 | 0.0656 | 0.1920 | 0.8884 | 0.0444 | 0.0055 | 0.3463 | 0.8219 | 0.9771 | 0.7619 | 0.5402 | 0.1093 | 2.42 |
| | LCD-UC | **0.0772** | **0.0677** | **0.1751** | **0.8992** | **0.0596** | **0.0130** | **0.2616** | **0.8413** | **0.9781** | **0.7625** | **0.5332** | **0.1206** | **1.00** |
| NGCF | Box | 0.1009 | 0.0974 | 0.2352 | 0.8549 | 0.2194 | 0.0606 | 0.1593 | 0.8177 | 0.9760 | 0.7604 | 0.5630 | 0.0712 | 2.75 |
| | LCD | 0.1025 | 0.0974 | 0.2304 | 0.8564 | 0.2283 | 0.0617 | 0.1488 | 0.8264 | **0.9791** | **0.7630** | 0.5632 | 0.0710 | 2.00 |
| | LCD-UC | **0.1065** | **0.1027** | **0.2252** | **0.8577** | **0.2632** | **0.0846** | **0.1173** | **0.8626** | 0.9781 | 0.7626 | 0.5612 | 0.0742 | 1.17 |
| LGCN | Box | 0.1028 | 0.1040 | 0.2563 | 0.8391 | 0.2542 | 0.0843 | 0.1277 | 0.8486 | 0.9783 | 0.7620 | 0.5635 | 0.0706 | 2.67 |
| | LCD | 0.1045 | 0.1043 | 0.2594 | 0.8320 | 0.2242 | 0.0725 | 0.1173 | 0.8591 | 0.9786 | 0.7621 | 0.5632 | 0.0711 | 2.33 |
| | LCD-UC | **0.1091** | **0.1071** | **0.2498** | **0.8399** | **0.2708** | **0.0894** | **0.1111** | **0.8668** | **0.9787** | **0.7625** | **0.5631** | **0.0718** | **1.00** |

we report $k = 20$ and similar results are observed when $k = 3$, $k = 5$ and $k = 10$.

To conduct a comprehensive evaluation of different methods on accuracy and diversity, we rank the results of these methods under different metrics and calculate the average rank for each method.

## 5.2 Effectiveness of LCD-UC (RQ1)

In this subsection, we examine how LCD-UC improves the performance of base models. We choose MF, NGCF and LGCN as the base models. Table 2 reports the experimental results and it is observed that: 1) Compared to the base model, LCD-UC can enhance recommendation accuracy, because box embeddings can represent the range of user interests and item features more precisely than point embeddings, which also shows the effectiveness of the hypercube similarity scoring function. 2) LCD-UC achieves better results than the base models in terms of diversity metrics, thanks to the range-based representation of boxes, the attention mechanism and the uncertainty masking mechanism. Thus, LCD-UC is demonstrated to be an effectiveness universal framework to enhance both the recommendation accuracy and diversity on different base models.

## 5.3 Overall Performance (RQ2)

In this subsection, we evaluate LCD-UC with existing recommendation methods. Specifically, we use NGCF, LGCN and MF as the base models of LCD-UC for MovieLens, KuaiRec and MIND, respectively, because these models get the best average rank in these datasets. We compare LCD-UC with the other baseline methods not included in Section 5.2. Note that we retune the parameter of LCD-UC on MovieLens to balance the performance on recommendation accuracy and diversity.

As illustrated in Figure 3, LCD-UC obtains the best average rank (i.e., the largest size of circle), showing that LCD-UC can make a good balance between accuracy and diversity. Specifically, LCD-UC achieves the state-of-the-art performance w.r.t. recommendation accuracy on all the datasets, and performs competitively in recommendation diversity. It is observed that PD-GAN and DGCN, which are designed for diversified recommendation, sacrifice the recommendation accuracy to obtain good diversity performance, and thus are poor in NDCG. In contrast, session-based methods like GRU4REC, HCUR and CBox4CR have comparable accuracy results, but are poor in ICSI.

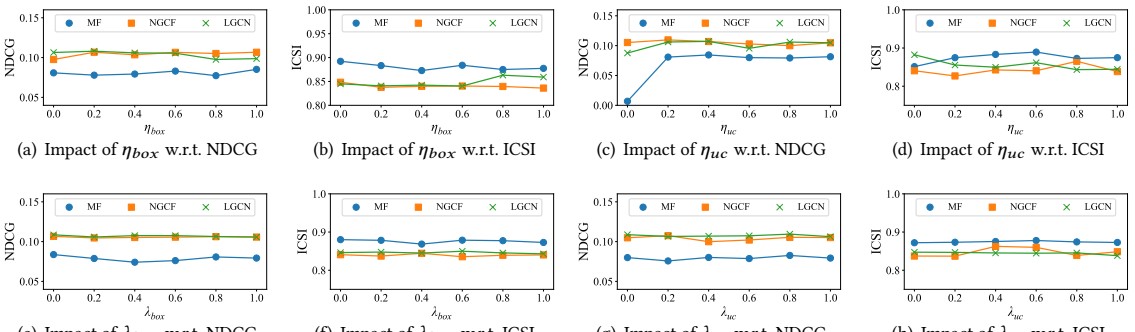

Figure 4: The experimental results of parameter sensitivity.

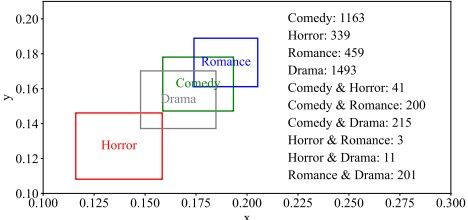

Figure 5: Visualizations of the average box of four categories on MovieLens. The box embeddings of different categories can essentially reflect their quantitative relationships.

## 5.4 Ablation Study (RQ3)

In this subsection, we conduct ablation studies to investigate the effectiveness of the user-item pairwise attention mechanism and the user uncertainty masking mechanism in LCD-UC. Specifically, we denote Box as the variant in which none of the two mechanisms are used, i.e., $s(u, i) = \sum_{k=1}^{d} h'^{(k)}_{u,i}$, and LCD as the variant in which the uncertainty masking mechanism is dropped.

Table 3 shows the experimental results. We can find that: 1) Even without the two mechanisms, Box is better than the base models (by comparing the results in Table 2 and Table 3). 2) LCD is superior to Box on average, showing the effectiveness of the attention mechanism. 3) LCD-UC obtains the best performance in almost all the cases. We attribute these results to the fact that the attention mechanism and the uncertainty masking mechanism can model personalized diversity needs for accurate and diversified recommendation.

## 5.5 Parameter Sensitivity (RQ4)

In this subsection, we analyze the sensitivity of hyper-parameters in LCD-UC. Specifically, we examine the impact of hyper-parameters $\eta_{box}$ and $\eta_{uc}$ in the box regularization loss and the mask regularization loss, respectively. We also investigate how different losses influence LCD-UC, i.e., the sensitivity of $\lambda_{box}$ and $\lambda_{uc}$. We report the NDCG and ICSI metrics on the MovieLens dataset.

As shown in Figure 4(a) and 4(b), different base models have varying sensitivity to $\eta_{box}$. For LGCN, we can observe that larger $\eta_{box}$ yields better diversity and worse accuracy, because bigger

boxes are easier to overlap and harder to distinguish with other boxes. From Figure 4(c) and 4(d), it is observed that setting $\eta_{uc}$ to $[0.6, 0.8]$ is good choices. Figure 4(e), 4(f), 4(g) and 4(h) show that LCD-UC is insensitive to $\lambda_{box}$ and $\lambda_{uc}$.

## 5.6 Box Visualization (RQ5)

In this subsection, we present a box visualization figure for better insight of the box embeddings learned by LCD-UC. We run LCD-UC with $d = 2$ to obtain 2D boxes. Figure 5 exhibits the visualization of the average box of items from four categories, together with the count of movies from those categories in the MovieLens dataset. We can observe that the overlapping relationships of the boxes roughly reflect the quantity relationships between different categories, which indicates that LCD-UC indeed has the capacity to learn interpretable box embeddings.

## 5.7 Online Performance (RQ6)

We deployed the LCD-UC model during the recall stage of an advertising system w.r.t. a video-watching platform with more than 400 million daily active users. The results of a 10-day AB test indicate that LCD-UC increases category diversity into the prerank stage by 0.685% and meanwhile advertising income by 0.812% when compared to the base vector model in the effective scenario. For a single recall model among various recall algorithms, this represents a significant improvement.

## 6 CONCLUSION

In this paper, we presented LCD-UC, a universal framework to learn box embeddings for users and items. The similarity between users and items were measured by a novel hypercube similarity scoring function. We also designed an attention mechanism and an uncertainty masking mechanism to achieve personalized diversified recommendation. Then, we proposed two new recommendation diversity evaluation metrics to resolve the limitation that existing diversity metrics failed to consider the frequency of item categories. The comprehensive experimental results empirically showed that LCD-UC was effective and can enhance diversity metrics with recommendation accuracy maintained. In the future, we plan to develop LCD-UC to represent each user and item with multiple hypercubes for better model flexibility and expressiveness.

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
