# OpenReview forum: "Enhancing Recommendation Accuracy and Diversity with Box Embedding: A Universal Framework"
_ACM.org/TheWebConf/2024/Conference — TheWebConf24_

### Official Review · Reviewer_uyRv · 2023-11-23

**Novelty:** 4
**Technical Quality:** 3

**Review:**

The paper proposes a new recommendation algorithm that utilizes box representations to increase diversity in recommendation systems. The idea is to start with a black-box single-point embedding of users and items in a common latent space and expand those embeddings to boxes. Intersection of user and item boxes indicates potential for recommendations, and since the user and item boxes span a larger volume, there is a larger possibility for diverse recommendations. The proposed method discusses how to learn the box representations so as to minimize different types of losses. In addition to the box method, the paper suggests two new measures to evaluate recommendations that capture better the notion of diversity.

The method is conceptually interesting and the empirical evaluation shows that it outperforms a large number of baselines.

However, in my view, the paper has some important weaknesses. Notably, there is not sufficient justification of all the design choices. The paper reads as a sequence of ad hoc steps without properly explaining the motivation for each steps. Reads the paper leaves many questions unanswered for the reader. For example:
1. (perhaps more important) The volume of high-dimensional boxes increases exponentially with the dimension. What is the impact of this fact to the recommendation algorithm?
2. Similar to the previous point, a box seem to include a lot of "empty" space. What about other alternative representations that do not have this drawback, for example, a point distribution?
3. In terms of motivation, the author discuss how price ranges or price fluctuations motivate the use of boxes. In my view, this is not consistent with other motivating discussion where the boxes are used to capture different interests of the user and enhance diversity.
4. I can agree that boxes can help capturing more broad interests of users, and thus, boxes can be used to represent users more accurately. But why do we need boxes to represent items? Arguably, items are about a single thing, and then a point representation is sufficient and more accurate.
5. The paper talks about box center and box offset. Is the offset the same for all dimensions (making it a hyper-square)? The paper does not specify.
6. How is the MLP used to find the box center and offset?
7. How are the dimensions d and d' decided?
8. There are a lot of technical details, e.g., about the Gumbel distribution or about the uncertainty masking mechanism, which are not properly discussed.

**Questions:**

I would appreciate to answer the questions above, except perhaps the last one, which is not really a question.

**Ethics Review Description:**

To the best of my understanding, there are no ethics review issues.

**Reviewer Confidence:**

2: The reviewer is willing to defend the evaluation, but it is likely that the reviewer did not understand parts of the paper

**Scope:**

3: The work is somewhat relevant to the Web and to the track, and is of narrow interest to a sub-community

---

### Official Review · Reviewer_CW9W · 2023-11-24

**Novelty:** 6
**Technical Quality:** 6

**Review:**

The paper introduces a novel method to employ the box embedding technique for recommendation. And a well-designed similarity function is proposed to estimate the preference of a user to an item, which is demonstrated to be effective in modeling users’ preferences. To enhance the diversity, the authors propose the Uncertainty Masking mechanism to mask some information on user embedding. Furthermore, two new metrics are proposed for evaluating the diversity. Experimental analysis is given to demonstrate the effectiveness of the proposed method.

### Strengths

1. The motivation to employ box embedding to improve the diversity is promising and interesting. And the designed method is very interpretable, which also demonstrates great effectiveness in experiments.

2. The similarity function is well-designed and effectively captures the underlying relationships between user and item, contributing to the overall accuracy.

3. The author carefully considered the shortcomings of existing diversity evaluation metrics, and meticulously designed two more rational metrics, making a significant contribution to the advancement of the field.


### Weaknesses

1. Some experimental settings and results are not described clearly. Specific problems refer to Question part.

2. Figure 3 appears unclear, and readers may easily confuse the legend with data points, leading to potential misunderstandings.

**Questions:**

### Questions

1. Why not report the retuned metrics on MovieLens in Table2? And what is the base model in Figure 3?

2. According to Table3, the removal of UC would comprise the accuracy. Since that the UC mechanism masks some information about the user, why does it still benefit the performance of accuracy?

3. The proposed UC mechanism seems like a universal strategy for improving diversity. What about to combine it with baseline methods, such as the MF or GRU4Rec?

4. Can you explain why “LCD-UC is insensitive to 𝜆𝑏𝑜𝑥 and 𝜆𝑢𝑐”?

**Reviewer Confidence:**

3: The reviewer is confident but not certain that the evaluation is correct

**Scope:**

4: The work is relevant to the Web and to the track, and is of broad interest to the community

---

### Official Review · Reviewer_p4hk · 2023-11-24

**Novelty:** 3
**Technical Quality:** 3

**Review:**

The manuscript proposes a LCD-UC framework to enhance the accuracy and diversity of traditional recommender systems using box embedding and an uncertainty masking mechanism. While the model demonstrates some universality and effectiveness, several concerns are noted:
1. The overall performance comparison lacks clear indications of improvements, making it difficult to assess the actual enhancement brought by the LCD-UC model.
2. The primary base models, NGCF and LGCN, are both neural graph-based, with LGCN being a simplified version of NGCF. This narrow selection does not sufficiently demonstrate the universality of LCD-UC.
3. The authors' motivation to use box embedding for enhancing diversity is not fully convincing, especially since box embedding lacks the repulsive property like DPP. It raises the question of whether other representation learning methods, coupled with user-item pairwise attention and user uncertainty masking mechanisms, could also improve diversity.
4. A key motivation for using box embedding in the proposed model is to enhance diversity. However, there seems to be a contradiction in this approach, as existing methods utilizing box embedding, such as CBox4CR, have not demonstrated particularly strong results in enhancing diversity. This apparent contradiction between the intended purpose and the observed outcomes in similar methodologies highlights a critical area that needs addressing or clarification in the work.
5. This work closely resembles another paper Lang Mei et al. "Learning Probabilistic Box Embeddings for Effective and Efficient
Ranking", even down to the settings of numerous parameters. This further reduces my regard for the originality and novelty of this manuscript. It should be clearly mentioned which parts are referred to existing works.
6. Considering the multiple existing box embedding-based recommendation models, the introduction of a new similarity scoring function and uncertainty masking mechanism seems to offer marginal contributions.
7. The use of examples to demonstrate the superiority of ICSI and ILCS over CC and ILD is not entirely convincing. Moreover, the absence of common diversity metrics like category coverage (CC, termed as GC in the work) and intra-list diversity (ILD) in the evaluation weakens the argument. The lack of comparison with harmonic metrics like the F-score for balancing accuracy and diversity further limits the assessment of the model's trade-off capabilities.
8. Figure 6 shows minimal performance trend changes for ICSI and ILCS under varying parameters, raising questions about the significance of these parameters, the design of ICSI and ILCS, or the actual impact of box embedding on diversity enhancement.
9. The use of different multi-layer perceptrons (MLPs) to learn representations suggests that performance improvements might be attributed to additional parameters and longer training epochs rather than the model's intrinsic qualities. The criteria for experimental result selection (best accuracy or best diversity?), the application of an early stop strategy, and the number of training epochs are not specified, leaving uncertainties about the robustness of the findings.
10. The manuscript does not provide details on the implementation of baselines. In addition, the proposed LCD-UC model is characterized by multiple parameters that require manual tuning. This aspect could hinder its ease of use and raises concerns that the reported improvements in accuracy and diversity might be largely attributable to manual parameter adjustments.

**Questions:**

Please refer to issues in review.

**Reviewer Confidence:**

4: The reviewer is certain that the evaluation is correct and very familiar with the relevant literature

**Scope:**

4: The work is relevant to the Web and to the track, and is of broad interest to the community

---

### Official Review · Reviewer_AUCB · 2023-11-24

**Novelty:** 6
**Technical Quality:** 5

**Review:**

This paper proposed a framework named LCD-UC to enhance accuracy and diversity in the recommendation, aiming at resolving low model flexibility and expressiveness of the current point embedding-based models. Specifically, a box-embedding-based framework LCD is proposed to encode the user and item representations into hypercubes. Then, a hypercube similarity scoring function is designed to measure the similarity between users and items. Besides, an attention mechanism and an uncertainty masking mechanism is proposed to achieve personalized diversified recommendation. Finally, two metrics are proposed to evaluate the diversity of the recommendation.
Pos:
1. The motivation is practical in industrial recommendation systems and clearly described.
2. The methods are well-described and seem easy to apply.
3. Extensive experiments are conducted and the results are thoroughly analyzed.

Cons:
1. How to guarantee accuracy theoretically when replacing the "meet a certain value"(with point embedding) with “meet a certain range” (with box embedding).
2. Conventional methods for diversity such MRR or DPP-based methods are not compared in the experiments.
3. There are still two hyper-parameters to control the diversity and accuracy, which makes it hard to demonstrate the flexibility and expressiveness of the box embedding.

**Questions:**

Please refer to the Cons.

**Ethics Review Description:**

There is no ethics issue.

**Reviewer Confidence:**

3: The reviewer is confident but not certain that the evaluation is correct

**Scope:**

4: The work is relevant to the Web and to the track, and is of broad interest to the community

---

### Official Review · Reviewer_aQnb · 2023-11-27

**Novelty:** 6
**Technical Quality:** 6

**Review:**

The paper is well written and addresses a common need in the recommendation space in terms of diversification. Though it is not clear how big a problem it is as the authors do not give specific evidence of it. Nonetheless, the authors do show they understand the nuances of the space in terms of balancing personalization, novelty, relevance and their “cocoons”.  They also identify an algorithmic cause, in terms of point embeddings failing to describe a range of user preferences.  And their solution is a generally interesting and innovative one, using hypercubes, and box embeddings that outline a range of preferences. It is these embeddings and approach that is more interesting than the issue of diversity per se. The references are rather diverse and quite interesting in terms of their breadth, which adds to the conviction that the work is well-founded and applicable to various use cases.

The authors introduce a List-Check-Decide framework with Uncertainty masking for their diversity mechanism.  Though they use the abbreviated term “LCD-UC” in their abstract, they not define it until line 150. This should be done at the initial mention for clarity.

To utilize box embeddings, as opposed to point ones, they developed a new hypercube similarity scoring function.  They also employ a “Uncertainty Masking Mechanism”, which reduces the importance of some user preferences so more items can be matched with the user.  This approach is interesting in itself for personalizing recommendations, where you need a mechanism to relax preferences deduced from previous behavior when the context of the query has overriding elements.

Two new diversity metrics are introduced and explained with easy to understand examples.  There is “Item Category’s Simpson’s Index” and “Intra-List Category Similarity”. They compare them to Genre Coverage and Intra-List Distance. From a practical perspective, it would have been good not just to look at sample performance of these metrics but also discuss their general stability, sensitivity to noise, and other “real world” elements that will not come from evaluation against MovieLens, for example.

The experimental results were quite positive and were encouraging for the box embedding approach.  It was appreciated to have a selection of 9 baseline methods, but the selection of MovieLens, KuaiRec and MIND did not provide extensive breadth in the recommendation space.  The latter do not capture dynamic ecommerce dynamics that you have on the Internet.

The approach was tested online on a platform with 400 million visitors to show the real world effectiveness of the methods. This testing differentiates the paper and provides a step-change in the confidence of the work and its utility. Unfortunately the test is not discussed in detail.

**Questions:**

Would it be possible to discuss the online test in more detail. Specifically, any mechanisms or explanations on the increase in ad revenue?
How general would you expect the concept of box embeddings to be?

**Ethics Review Description:**

No issues

**Reviewer Confidence:**

3: The reviewer is confident but not certain that the evaluation is correct

**Scope:**

4: The work is relevant to the Web and to the track, and is of broad interest to the community

---

### Decision · Program_Chairs · 2024-01-22

**Decision:**

Accept

**Comment:**

The paper suggests a recommendation algorithm employing box representations to enhance diversity in recommendation systems. Reviewers have highlighted issues such as insufficient rationale for the design choices, limited experimentation and comparison with key related works, and an absence of theoretical result analysis. The paper's evaluation is currently at the borderline.